# Energy Availability and Body Composition in Professional Athletes: Two Sides of the Same Coin

**DOI:** 10.3390/nu16203507

**Published:** 2024-10-16

**Authors:** Roberto Palazzo, Tommaso Parisi, Sara Rosa, Marco Corsi, Edoardo Falconi, Laura Stefani

**Affiliations:** Sports Medicine Centre, University of Florence, 50121 Firenze, Italy; roberto.palazzo@unifi.it (R.P.); tommaso.parisi@edu.unifi.it (T.P.); sara.rosa@unifi.it (S.R.); marco.corsi@unifi.it (M.C.); edoardo.falconi@unifi.it (E.F.)

**Keywords:** energy availability, energy intake, energy expenditure, body composition, rugby players, endurance athletes

## Abstract

Background/Objectives: Energy availability (EA) is essential for maintaining physiological functions, significantly influencing athletes’ health and performance. Nutritional behaviors, however, vary across sports. This study aims to assess EA levels in athletes from different disciplines, focusing on the relationship between EA and body composition in endurance athletes compared to rugby players. Methods: This study involved 18 endurance athletes (15 men, 3 women) and 36 rugby players (all men). Data were gathered through interviews, questionnaires, and bioimpedance analysis. Energy intake (EI) was measured with a 24 h dietary recall, and exercise energy expenditure (EEE) was calculated using the IPAQ questionnaire. EA was calculated as EA = (EI − EEE)/fat-free mass (FFM), with results categorized into clinical, subclinical, and optimal ranges. Results: The endurance group had a lower average FFM (57.81 kg) compared to the rugby players (67.61 kg). EA was also significantly lower in endurance athletes (11.72 kcal/kg FFM) than in rugby players (35.44 kcal/kg FFM). Endurance athletes showed more restrictive nutritional behavior with lower EI and higher EEE, but both groups maintained body composition within normal ranges. Conclusions: Endurance athletes exhibit greater nutritional restrictions compared to rugby players, though their body composition remains healthy. Further research is required to investigate the long-term effects of low EA on performance, injury risk, and potential impairment when EA falls below the optimal threshold of 45 kcal/kg FFM/day.

## 1. Introduction

Energy availability (EA) is an important determinant of the nutritional status of the general population and also of athletes. By definition, it is specifically referred to as the amount of energy available for physiological functions after accounting for exercise energy expenditure. It cannot, therefore, merely link to caloric intake versus expenditure. The concept of energy balance (EB) is, on the contrary, used to explore the effects of insufficient energy intake (EI) [1]. The critical aspect of the nutritional equilibrium in athletes is currently of interest in sports medicine. This is particularly due to the complex management required to balance caloric intake with the energy expended during training [2]. In the presence of energy deficiency, negative impacts in some districts have been described [3]. A discrepancy has been found among competitive athletes, characterized by low EA levels, which can have potential health and sports-related consequences, such as reduced performance and increased risk of injury [4]. Recent studies have demonstrated a prevalence around the 25% of low EA in various sports [5]. LEA has been described in male road cyclists [6] and elite distance athletes [7].

A more accurate evaluation of the effective energy necessity can be expressed as “Relative Energy Deficiency in Sport” syndrome [4] as the face of insufficient EI consumption to support daily energy expenditure, resulting in the state of being effective, low energy availability (LEA), and decrements in sports performance.

Following the current definition, LEA is commonly defined as EA <  30 kcal/kg FFM. This can represent a clinical threshold for LEA [8], potentially inducing a negative impact on an athlete’s performance. This definition can, therefore, effectively support the correct interpretation of optimal vs. suboptimal levels of EA in athletes.

This issue seems to be associated with poor knowledge in this field [9] among athletes who tend to consider a low Body Mass Index (BMI) as a key component for optimal performance. It is, in fact, common to limit daily caloric intake, especially by reducing carbohydrates, thereby increasing the risk of developing a state of low EA. In addition to limiting caloric intake, these sports involve very high training volumes, which further widen the gap between energy intake and calories burned through sports activity. This behavior is often associated with a kind of dependence on maintaining this incorrect approach [10]. On the other hand, in sports, as team or singular sports, such as particularly those where there is a combination of an endurance component and physical strength, like endurance athletes, such as swimmers, cyclists, or runners, a low EA seems to be prevalent. This aspect has not been largely evaluated, especially among sports with long-distance training (cyclists, runners, swimmers) who need to maintain a constant performance for long periods if compared to team sports featuring combined elements (aerobic/anaerobic metabolism), like rugby. The first group can potentially be considered a category more exposed to the risk of low EA. Here, an incorrect eating behavior is often associated with believing that a reduction in calorie intake can guarantee a high performance in terms of body weight control. No large data are available on the comparison of these two types of sports in terms of LEA presence.

The current literature does not report a specific trend among these different types of sports. Therefore, considering the different kind of training, the objective of this study is to assess EA levels and body composition in athletes practicing in different sport disciplines in terms of aerobic and anaerobic intensity, however, with high energy demand, and to evaluate any differences in the incidence of low energy availability or increased risk of developing it based on the varying athletic demands of the sport practiced.

## 2. Materials and Methods

Population Studied:

The study was conducted with two groups of athletes. The first group (18 subjects, including 15 men and 3 women) consisted of endurance athletes practicing disciplines such as long-distance running, cycling, and swimming. They had an average age of 47.61 ± 11.37 years, a weight of 74.47 ± 8.75 kg, and a height of 175.78 ± 8.57 cm. Their BMI was within the normal range at 23.92 ± 1.67 kg/m^2^. The second group comprised a team of athletes, including 36 male rugby players, with an average weight of 86.50 ± 7.07 kg, an average height of 181.83 ± 5.66 cm, and an average BMI of 26.15 ± 1.62 kg/m^2^. The athletes were engaged in regular training as normally followed in the seasonal time. They were from teams engaged in seasonal competitive games. They can be considered professional, despite non-elite athletes. For this reason, no specific details regarding the intensity of the exercises performed, the speed of the run, or other specific characteristics of the training are available or collected in the present study. The endurance athletes (E) trained for almost 3 h per session at least 3–4 days a week, with swimmers training every day for at least 2 h. The rugby players (R) trained for at least 3 h, 3 times a week, and played a game on the weekend. Each participant provided written informed consent for data processing and participation in the evaluation. However, since they were regularly monitored at the Sports Medicine Center of the University of Florence, approval from an ethics committee was not necessary. Data are available on the Sports Medicine website on behalf of the corresponding author.

Methods:

The LEAF-Q (Low Energy Availability in Females Questionnaire) was used to estimate the risk of LEA [11] in the small sample of females. For the rest of the male sample, the LEAM-Q was used, as some studies have adopted it [12].

### 2.1. Face-to-Face Interview

During the interview, a 24 h recall was conducted to evaluate the participants’ energy intake (EI). Each subject was asked to detail all meals consumed the previous day, including snacks, condiments, supplements, and food brands, and an estimate of habitual water consumption. Athletes were also asked to provide as precise an indication of food portion sizes as possible. When they were unsure of the weights, a food atlas was used to identify the quantity consumed for each food. After obtaining a detailed description of the daily food intake, the total daily calories were calculated to determine the energy intake (EI) of each athlete. The method suffered, however, from individual interpretation and potential inaccuracies. Despite this, the application of the face-to-face method in the present study was due to its importance to prefer, in an ambulatorial setting, a more colloquial scheme. During the face-to-face interview, athletes were also asked to describe in detail the types and duration of their workouts, providing information on the volume and intensity of the sessions. To find the average daily training time, the total weekly training hours were divided by seven.

### 2.2. IPAQ Questionnaire

To assess exercise energy expenditure (EEE), the IPAQ questionnaire was administered [13,14,15]. This questionnaire provides a result in METs (metabolic equivalents) consumed per minute based on the previous week’s workouts. This value was then converted into METs consumed per hour per training day by dividing by 60 and then by the total weekly training hours.

Once these data were obtained, EEE was calculated. METs are expressed in $\frac{Kcal}{h*weight}$ using the formula Kcal=MET∗h∗weight, where METs are those obtained from the IPAQ converted into hours, h represents the daily training hours, weight is the athlete’s weight, and the EEE of each participant was calculated [16].

The bioelectrical impedance analysis provided data on body composition and hydration status. Detailed aspects such as FFM (fat-free mass), which is of interest for calculating athletes’ energy availability using the formula $EA=\frac{EI-EEE}{FFM}$, ASMM (appendicular skeletal muscle mass) using the Janssen formula, FM (fat mass), and phase angles were assessed [17,18].

Energy availability was determined using the following value ranges: clinical state if <30 kcal/kg FFM for both sexes, subclinical state for values of 30–40 kcal/kg FFM for men or 30–45 kcal/kg FFM for women, and optimal state for values ≥40 kcal/kg FFM for men and ≥45 kcal/kg FFM for women.

### 2.3. Body Composition Analysis

A bioimpedance analysis was selected to assess body composition. The BIA measurements were conducted under standardized conditions, early in the morning, after 10 min of rest, and inside a room at normal and constant temperature and humidity. The athletes were asked not to drink alcoholic beverages in the 12 h before the exam, not to perform intense physical activity in the 12 h before the exam, not to use a sauna in the 12 h before the exam, and not to drink liquids for 2 h before the exam time.

The bioelectrical parameters of resistance (R) and reactance (Xc) were obtained using a BIA 101 Anniversary Sport Edition analyzer (Akern Srl, Florence, Italy), which generates an alternating sinusoidal current of 400 mA at 50 kHz (±0.1%). Prior to each assessment, this instrument was calibrated with a known impedance circuit supplied by the manufacturer.

The measurements were performed following the guidelines, with arms and legs slightly apart to prevent contact with the body. The readings were taken after a 5 min stabilization period, during which the participants remained still to ensure a uniform distribution of body fluids. The injector electrodes were placed on the dorsal surface of their right hand (near the third metacarpophalangeal joint) and right foot (near the third metatarsophalangeal joint). The sensing electrodes were positioned approximately 5 cm away from the injector to avoid interaction between the electric fields and to prevent overestimation of the impedance values.

Impedance (Z) was determined as (R^2^ + Xc^2^)^1/2^ and the phase angle (PhA) as tan^−1^ (Xc/R·180°/π). R, Xc, and Z were adjusted for height (R/H, Xc/H, Z/H). According to conventional BIVA, Z/H is inversely related to total body water (TBW) [19]. Conversely, vector direction indicates cellular health and cell membrane integrity and is inversely related to the extracellular/intracellular water (ECW/ICW) ratio [20]. All interpretations should consider both Z/H and PhA in combination with the vector position on the Resistance–Reactance (RXc) graph [21]. On this graph, shifts in vectors parallel to the major ellipse axis reflect differences in tissue hydration (a longer vector indicates less fluid, while a shorter vector suggests more body fluids). Shifts in the vector parallel to the minor axis of the ellipses indicate changes in cell mass and the ECW/ICW ratio (a shift to the left suggests an increase in cell mass and a reduction in the ECW/ICW ratio, while a shift to the right indicates a decrease in cell mass and an increase in the ECW/ICW ratio [22]).

### 2.4. Statistical Analysis

All data were expressed as mean ± SD. The general data and the data from BIA were compared by Student’s *t*-test for paired data. The significance of the differences between the results of the two groups of athletes was evaluated using a two-tailed *t*-test pointed at *p* < 0.05. The CI was also calculated. The data obtained from the questionnaires (IPAQ-LEAF-Q, LEAM-Q EA, EE) were analyzed using a Google Form system (https://www.google.com/forms/about/, accesses on 12 October 2024) and expressed in % on the basis of the value classification reported in the literature.

## 3. Results

The body composition was assessed by BIVA (bioelectrical vector analysis). The values of fat-free mass (FFM), skeletal muscle mass (SMM) estimated by the Janssen formula, and fat mass (FM) were measured for both groups of athletes.

E athletes had an average fat-free mass (FFM) of 57.81 ± 7.62 kg, while R players had a higher average of 67.61 ± 5.01 kg.

The percentage of appendicular skeletal muscle mass (ASMM) was similar between the two groups: the first group had an average of 30.01 ± 6.38 kg, and the second group had an average of 27.26 ± 4.70 kg.

In E athletes, the FM value was 22.07 ± 4.77%, and in rugby players, the value was 21.66 ± 4.34%, which are not different. On the contrary, the FFM values resulted in significantly different values between the two groups, despite being similar for ASMM (Janssen), as reported in Table 1. The average phase angles for the two groups were 6.37 ± 0.54 for the E athletes and 7.13 ± 0.40 for the R players. The difference between these values was statistically significant, despite being within the normal range. From the IPAQ questionnaire, the E athletes showed an EE of 6460 ± 1565.41 MET, while the R players had 4152.14 ± 2447.35 MET. The training level for E athletes was between 10 and 14 h per week, with an average of 12.39 ± 1.50 h per week, which is equivalent to 1.77 ± 0.21 h of daily training. The R players, on the other hand, reported to train at least three times a week, totaling about 7 h per week, which is equivalent to an average of 1 h of daily training.

Based on the data and face-to-face interviews with the athletes, the EI, EEE, and EA values for each athlete were calculated. For the E group, the EI was on average 1796.89 ± 405.46 kcal per day, the EEE was 1144.41 ± 311.32 kcal per daily training session, and the EA was 11.72 ± 9.05 kcal/kg FFM. Rugby players (R) had an average EI of 3237.28 ± 158.80 kcal per day, an average EEE of 839.52 ± 463.78 kcal per daily training session, and an average EA of 35.44 ± 6.63 kcal/kg FFM. (Table 1).

Examining the data for each athlete, 17 endurance athletes were in a clinical state of energy availability, and one athlete was in a subclinical state. No athlete in the first group had an optimal energy availability level. As for the rugby players, 7 were in a clinical state of energy availability, 22 were in a subclinical state, and 7 were in an optimal state. The global EA of the two groups of athletes classified as clinical, suboptimal, and optimal are summarized in Figure 1A,B. EI, EEE, and EA levels in the two groups of athletes are shown in Figure 2.

## 4. Discussion

EA is an important aspect of the management of nutritional status in athletes. The long-term negative effects of LEA are largely described, especially in hormonal health, bone health, and injury risk [23]. The discrepancies in the presence of a high energy requirement are not often evident; they remain asymptomatic and emerge only in the presence of a large and in-depth investigation. It is well recognized that LEA in athletes may face negative consequences across various physiological systems, including endocrine, cardiovascular, immune, metabolic, reproductive, and gastrointestinal [24]. From the literature, it seems that some sports, like sports with long-distance training, including endurance sports, are more exposed to this risk [11]. A low EA, due to a low EI, is often associated with an expectation of higher performance. From the present investigation, it seems that rugby has a lower risk of LEA; however, this study cannot investigate the complex mechanism that supports this hypothesis, and that is partially due to the missing specific diet and metabolic needs. In addition, the face-to-face interview potentially obscured some aspects. The data collection and analysis reveal, according to the literature [11,25], that the athletes in the two groups had different physical characteristics based on the type of sport practiced and different EAs. Endurance athletes had a lower weight compared to rugby players but had a similar amount of skeletal muscle mass and fat mass relative to their body weights. There is literature available on this aspect in rugby players, with particular attention on impaired sleep quality [26]. This is one of the physiological dysfunctions associated with LEA that induces more attention on the nutritional behavior. This is also evident when analyzing the average BMI values of the two groups, which were significantly higher for the rugby players than for the endurance athletes. These physical characteristics reflect the type of sport practiced; rugby players had higher body masses compared to endurance athletes. Endurance athletes had a very high training volume, reaching up to 14 h per week. Despite the significant physical effort required, the caloric intake from their diet was particularly low, sometimes as low as 1221 calories per day. The caloric intake was sufficient, for most athletes, only to sustain their basal metabolic rate. During interviews, the athletes often mentioned that nutritionists or professionals in the field did not follow them and that the choice to limit their caloric intake was voluntary, especially regarding carbohydrates. They believed that maintaining a low body weight would ensure performance regardless of the competitive preparation period. This significant gap between caloric intake and the energy expended during training results in very low available energy for the normal performance of all physiological functions. All athletes in this group were in a clinical state of low energy availability, with values significantly below the clinical threshold. Only one athlete fell into the subclinical range, and none were in the optimal range. The evaluation and the data reported are representative of the training days. This is particularly important as the expression of specific behavior in a training performance session. In this context, specific nutritional and professional support by nutritionists seems to be desirable. This kind of figure is not often present among professional athletes, while it is used in elite categories. A nutritionist could enhance the importance of maintaining a correct intake of the percentage of macronutrients and EI recommended: at least 45 to 65% from carbohydrates, 15 to 25% from protein, and 20 to 35% from fat [27]. This is recommended to avoid a lack of sufficient EI and to compromise athletic performance [28]. For the rugby athletes, the situation was significantly different. The training volume for this group was much lower compared to the endurance athletes, totaling about 7 h of training per week. Consequently, the caloric expenditure from the exercise was lower. This was associated with a much richer diet compared to the endurance athletes. Rugby players, needing to maintain a higher body mass and an important strength component to be competitive, had to maintain a higher caloric intake, with almost none reporting a dietary energy intake below 3000 calories per day. As a result, their energy availability values were significantly higher. Only 7 athletes were in a clinical state of energy availability, 22 were in a subclinical condition, and 7 were in an optimal state. Comparing the results obtained between the two groups of athletes, those practicing endurance sports had a significantly higher incidence of low energy availability; 94% of them were in clinical condition. In contrast, only 19% of rugby athletes were in a state of clinically low energy availability. Athletes in both groups had phase angles within the normal range, indicating good hydration status and healthy cell membrane structure. This study has some limits. Among them are the relatively small sample of the group investigated and, additionally, the disparity between the number of male and female participants. This limits substantially the eventual generalization of the conclusion, not indicative of gender differences. Another important aspect is the absence of an analysis of micronutrient intake, which can be significant in cases of low energy availability. A larger, more balanced sample will be determinant for robust statistical analysis and broader applicability of the results. Further studies will be necessary to implement these aspects.

## 5. Conclusions

According to the current literature and the data obtained, endurance sports appear to be more prone to issues of low energy availability compared to mixed sports with aerobic and strength components, such as rugby. Despite all the athletes investigated engaging in high-intensity training, rugby players seem to need to maintain a higher-than-average body weight. Endurance athletes, who are dedicated to long-term sports activities, often consciously limit their energy intake, sometimes engaging in disordered eating behaviors. They may consume a caloric amount suitable for a sedentary person in an attempt to appear leaner, weigh less, and gain a perceived advantage in long-distance races. This can have significant health consequences, as the body may not have sufficient energy to maintain essential physiological functions, leading to the suppression of some functions to preserve others. Over time, this dysregulation of energy balance could negatively affect health, including cardiac function, recovery from training, and athletic performance, and increase the risk of injuries. Regarding rugby, considered a mixed-team sport that includes endurance, strength, and muscle explosiveness components, it is not exempt from low energy availability issues, but it is less impactful compared to pure-endurance sports. For this athlete type, there seems to be greater awareness of the importance of dietary intake, driven by the need to maintain a physique and strength to remain competitive against players from other teams.

This study concludes by highlighting the importance for athletes, especially those practicing endurance activities, to be followed by a nutritionist or a professional in the field of nutrition and to be sensitized to the relevance of nutrition in sports. Managing this aspect autonomously, due to ignorance or false personal beliefs, risks significantly compromising their health. Relying on external support for dietary management will allow athletes to adjust their caloric and nutrient needs according to the volume and intensity of training, ensuring better recovery, maximizing performance, and improving or maintaining their health. Additionally, greater awareness among athletes about the role of nutrition will reduce the risk of developing disordered eating behaviors or actual eating disorders.

### Limitations and Future Perspectives

The main limitation of this study concerns the determination of athletes’ energy intakes. These were calculated using the 24 h recall method, which evaluates the subject’s diet over a very short period. Additionally, there may be a recall bias, where participants may not remember to mention some foods consumed, leading to an underestimation of the calories consumed.

A second limitation of this type of study lies in the lack of agreement in the scientific community on the method for quantifying values to calculate energy availability. Consequently, results can vary from study to study depending on the methodologies and tools used to identify energy intakes and calories burned through training. Identifying a validated method to calculate energy availability will, over time, allow for easier and more precise comparisons of results and conclusions from studies on this topic, increasing the possibilities of mitigating the problem and the consequences of low energy availability in sports.

## Figures and Tables

**Figure 1 nutrients-16-03507-f001:**
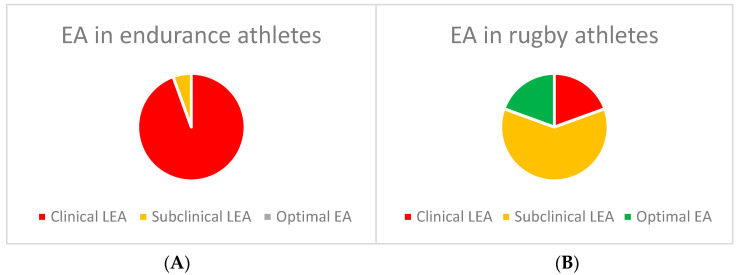
(**A**,**B**) representing the EA levels in endurance athletes (E) and in rugby players (R), respectively. In E athletes, no optimal EA was observed, while in R players, the majority were within the subclinical and optimal range.

**Figure 2 nutrients-16-03507-f002:**
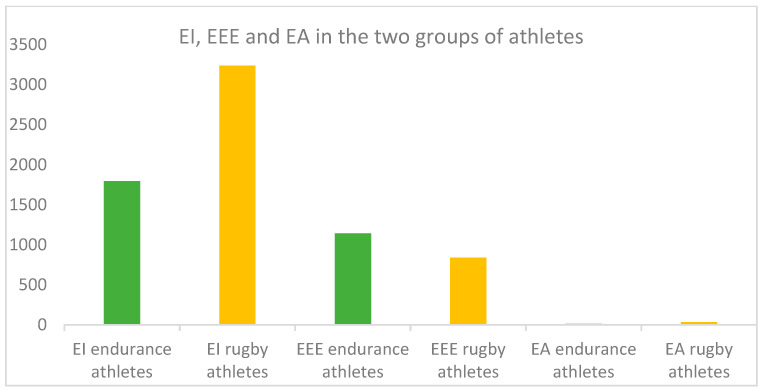
Comparison of average EI, EEE, and EA levels in the two groups of athletes.

**Table 1 nutrients-16-03507-t001:** Nutritional and anthropometric parameters of E and R athletes. Legend: EI: Energy Intake; EEE: Energy Expenditure; EA: Energy Availability; FM: Fat Mass; FFM: Free-Fat Mass; ASMM: Appendicular Skeletal Muscle Mass.

	Rugby Players						Endurance Athletes					
	t	*df*	Mean Difference	95% Confidence Interval of the Difference			t	*df*	Mean Difference	95% Confidence Interval of the Difference		
				Lower	Upper					Lower	Upper	*p* Value
EI	122.3	35	3237.28	3183.55	3291.01	EI	18.8	17	1796.88	1595.26	1998.51	*p* < 0.01
EEE	10.8	35	839.51	682.59	996.43	EEE	15.59	17	1144.41	989.59	1299.23	*p* = 0.01
EA	32.08	35	35.44	33.2	37.68	EA	5.49	17	11.72	7.21	16.22	*p* < 0.01
MB	122.18	35	1904.33	1872.70	1935.97	MB	48.40	17	1681.77	1608.46	1755.08	*p* < 0.01
FFM%	108.29	35	0.78	0.76	0.8	FFM%	49.40	17	0.76	0.73	0.80	*p* < 0.01
FM	29.94	35	0.21	0.20	0.23	FM	19.62	17	0.22	0.19	0.24	*p* < 0.01
ASMM (Kg)	35.13	35	27.22	25.64	28.79	ASMM (Kg)	19.65	17	30.01	26.78	33.23	*p* > 0.05
Phase Angle	76.68	35	7.08	6.89	7.27	Phase Angle	44.40	17	6.44	6.13	6.75	*p* < 0.01

## Data Availability

The data presented in this study are available on request from the corresponding author. The data are not publicly available due to privacy reasons.

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
