# Peer review of "Energy Availability and Body Composition in Professional Athletes: Two Sides of the Same Coin"

_nutrients, 2024, doi:10.3390/nu16203507_

Round 1
Reviewer 1 Report
Comments and Suggestions for Authors
The article addresses a very important and current issue of low energy availability (LEA) in sports. From a substantive perspective, the study provides valuable data, but some clarifications are needed, both in the methodology and in the discussion of the results.
Introduction
Lack of clarity in problem definition: The introduction highlights energy availability (EA) as an issue of concern, but it does not clearly distinguish between energy availability and energy balance, which are related but distinct concepts. The introduction should emphasize that EA refers specifically to the amount of energy available for physiological functions after accounting for exercise energy expenditure, not merely caloric intake versus expenditure. Inconsistent definition of athlete categories: The introduction describes discrepancies in energy availability across different sports, but it lacks clarity when explaining which sports tend to exhibit lower EA. For instance, the reference to sports with "a lower endurance component but greater physical strength" (line 41) should specify examples. Also, the observation that “mixed sports” may have different patterns in EA is vague. There is a lack of precise definition of the term Energy Availability (EA), especially the clinical thresholds for EA (e.g., <30 kcal/kg FFM as the threshold for LEA). Introducing this definition would allow for a better understanding of what EA values are optimal or harmful. The introduction does not refer to previous studies directly comparing energy availability across different types of sports, which would be beneficial for building the research context.
Methods
The study’s sample is relatively small, with 18 endurance athletes and 36 rugby players (line 51). While this may be acceptable for a preliminary study, it limits the generalizability of the results. Additionally, the disparity between the number of male and female participants (15 men vs. 3 women in the endurance group) is significant and should be addressed as a limitation since it could skew the results. Lack of details on the LEAF-Q adaptation: Although the LEAF-Q questionnaire is validated for women, it is mentioned that it was adapted for men (line 67), yet the methods for this adaptation are not clearly explained. This raises concerns about the appropriateness and accuracy of using a modified tool without a proper validation process for male athletes. This section should detail the specific changes made to the questionnaire for male athletes. Ambiguity in bioimpedance analysis methods: While the bioimpedance analysis (BIA) protocol is explained in detail (lines 103-127), it’s unclear whether the BIA measurements were conducted under standardized conditions (e.g., hydration status, time of day). Since BIA results can be affected by factors such as fluid intake and time of measurement, failure to control these could affect the reliability of body composition data. Although the authors mention that no ethics committee approval was necessary (lines 62-63), it would be beneficial to clarify the exact reasons for this exemption and whether informed consent alone suffices for studies of this nature. The statement could leave room for ethical concerns, especially regarding vulnerable groups (e.g., athletes with potential energy deficiencies). The article lacks a description of the standardization of measurement conditions for BIVA (e.g., time of day, hydration status). It is necessary to specify whether the tests were conducted in a specific physiological state (e.g., after overnight fasting), as this could significantly impact the results of the bioelectrical impedance analysis. The Janssen equation for assessing muscle mass is mentioned, but it is not explained why this particular formula was chosen and what its limitations are. In the case of EA (Energy Availability), it is not stated how energy intake (EI) and energy expenditure (EEE) were measured or estimated—whether through 24-hour recall interviews, dietary logs, or other tools. Subjective methods, such as interviews, may lead to inaccuracies, which should be noted.
Results
Some data are presented without full statistical information, e.g., when discussing differences between groups, it is not always clear which statistical tests were applied or what the significance levels (p-values) were. Indicating the tests (e.g., t-tests, ANOVA) and confidence intervals would enhance the credibility of the results. The data on calorie intake (e.g., the average EI for endurance athletes being 1796.89 kcal) suggest very low values, which is alarming. It would be important to clarify whether these values are representative of a full day or only for training days. Additionally, there is a lack of information on whether micronutrient intake, which can be significant in cases of low energy availability, was analyzed.
Discussion
The conclusions regarding rugby are too general. It seems that the lower risk of LEA in rugby players may be due to other factors, such as their diet and metabolic needs, but this issue is not sufficiently developed. The discussion does not directly reference existing research on LEA in rugby. It would be useful to analyze whether there are studies in the literature showing similar results. A more detailed analysis of the long-term effects of LEA is needed, especially in the context of hormonal health, bone health, and injury risk. The article mentions physiological dysfunctions but without detailed references to literature on the long-term consequences of LEA. There is also a lack of explanation regarding what specific interventions could help athletes improve their energy availability. Evidence-based recommendations regarding nutritional strategies for athletes in both groups should be included.
Conclusions
The conclusions could be more detailed, with specific recommendations for nutritional interventions and the need for educating athletes.
Author Response
The article addresses a very important and current issue of low energy availability (LEA) in sports. From a substantive perspective, the study provides valuable data, but some clarifications are needed, both in the methodology and in the discussion of the results.
We want to thank all the reviewers for the attention to our effort and to have appreciate our paper. We thank also to have suggested several important modifications addressed to improve ,to ameliorate and to valorise the message. All the changes have been made following your suggestions. We hope the changes are in agreement with your intention .
Introduction
Lack of clarity in problem definition: The introduction highlights energy availability (EA) as an issue of concern, but it does not clearly distinguish between energy availability and energy balance, which are related but distinct concepts. The introduction should emphasize that EA refers specifically to the amount of energy available for physiological functions after accounting for exercise energy expenditure, not merely caloric intake versus expenditure.
We are in agreement with you . The distinction of the two regimes and the EA definition has been made in the introduction session and the reference cited .
Inconsistent definition of athlete categories: The introduction describes discrepancies in energy availability across different sports, but it lacks clarity when explaining which sports tend to exhibit lower EA. For instance, the reference to sports with "a lower endurance component but greater physical strength" (line 41) should specify examples.
Thank you for this suggestion. The sentence has been added .
Also, the observation that “mixed sports” may have different patterns in EA is vague. There is a lack of precise definition of the term Energy Availability (EA), especially the clinical thresholds for EA (e.g., <30 kcal/kg FFM as the threshold for LEA).
Thank you for adding this suggestion . the definition has been insert
Introducing this definition would allow for a better understanding of what EA values are optimal or harmful. The introduction does not refer to previous studies directly comparing energy availability across different types of sports, which would be beneficial for building the research context.
Thank you for highlight this crucial aspect foundamental for the correct interpretation . We have added the definition
Methods
The study’s sample is relatively small, with 18 endurance athletes and 36 rugby players (line 51). While this may be acceptable for a preliminary study, it limits the generalizability of the results. Additionally, the disparity between the number of male and female participants (15 men vs. 3 women in the endurance group) is significant and should be addressed as a limitation since it could skew the results.
The limit of the study has been described
Lack of details on the LEAF-Q adaptation: Although the LEAF-Q questionnaire is validated for women, it is mentioned that it was adapted for men (line 67), yet the methods for this adaptation are not clearly explained. This raises concerns about the appropriateness and accuracy of using a modified tool without a proper validation process for male athletes. This section should detail the specific changes made to the questionnaire for male athletes.
the LEAF-Q is largely reported and used in literature among the female athletes as the most important validated method to estimate the low energy availability in endurance . In the present investigation, it has been used considering the presence of a small sample of female. At the same time the male version has been adopted for the rest of the athletes. Screening for Low Energy Availability in Male Athletes: Attempted Validation of LEAM-Q Bronwen Lundy,1,2 Monica K. Torstveit,3 Thomas B. Stenqvist,3 Louise M. Burke,2,* Ina Garthe,4 Gary J. Slater,5 Christian Ritz,6 and Anna K. Melin7 Nutrients. 2022 May; 14(9): 1873. Ambiguity in bioimpedance analysis methods: While the bioimpedance analysis (BIA) protocol is explained in detail (lines 103-127), it’s unclear whether the BIA measurements were conducted under standardized conditions (e.g., hydration status, time of day).
Thank you for this suggestion. Details have been added
Since BIA results can be affected by factors such as fluid intake and time of measurement, failure to control these could affect the reliability of body composition data. Although the authors mention that no ethics committee approval was necessary (lines 62-63), it would be beneficial to clarify the exact reasons for this exemption and whether informed consent alone suffices for studies of this nature. The statement could leave room for ethical concerns, especially regarding vulnerable groups (e.g., athletes with potential energy deficiencies).
Despite the study has been conduct in human , we notify that the exams performed are included in the global evaluation of the athletes in a Sports Medicine center dedicated to the athlete’s evaluation . Non invasive exame was performed , the BIA analysis that is a part this evaluation and the questionnaire represent an additional investigation made on the voluntary request of the athletes . They adhere spontaneously to this complete check up and for this reason sing an informal consent .
The article lacks a description of the standardization of measurement conditions for BIVA (e.g., time of day, hydration status). It is necessary to specify whether the tests were conducted in a specific physiological state (e.g., after overnight fasting), as this could significantly impact the results of the bioelectrical impedance analysis.
The details of the measurement by BIVA analisys are included in the text . In our center , as usually in others , the athletes receive the general indication before to be submitted to this eaxmination .
The Janssen equation for assessing muscle mass is mentioned, but it is not explained why this particular formula was chosen and what its limitations are.
As well know the Janssen equation is related to the evaluation of the Muscle Mass , in particular the appendicular Mass , therefore the significance is slightly different from FFM , however very important.
Thank you for this precisation.
In the case of EA (Energy Availability), it is not stated how energy intake (EI) and energy expenditure (EEE) were measured or estimated—whether through 24-hour recall interviews, dietary logs, or other tools. Subjective methods, such as interviews, may lead to inaccuracies, which should be noted.
We know this limit. We thank to have underlined . A sentence has been added to clarify the intention and the message
Results
Some data are presented without full statistical information, e.g., when discussing differences between groups, it is not always clear which statistical tests were applied or what the significance levels (p-values) were. Indicating the tests (e.g., t-tests, ANOVA) and confidence intervals would enhance the credibility of the results.
Statistical analysis is described in the specif session . The CI has been added . The table has been completely modifyed. Thank you
The data on calorie intake (e.g., the average EI for endurance athletes being 1796.89 kcal) suggest very low values, which is alarming. It would be important to clarify whether these values are representative of a full day or only for training days.
We are in agreement with you . A more emphasis of the data found has been described .
Additionally, there is a lack of information on whether micronutrient intake, which can be significant in cases of low energy availability, was analyzed.
This important aspect has been added in the limit of the study
Discussion
The conclusions regarding rugby are too general. It seems that the lower risk of LEA in rugby players may be due to other factors, such as their diet and metabolic needs, but this issue is not sufficiently developed.
The discussion does not directly reference existing research on LEA in rugby. It would be useful to analyze whether there are studies in the literature showing similar results. A more detailed analysis of the long-term effects of LEA is needed, especially in the context of hormonal health, bone health, and injury risk.
The article mentions physiological dysfunctions but without detailed references to literature on the long-term consequences of LEA. There is also a lack of explanation regarding what specific interventions could help athletes improve their energy availability. Evidence-based recommendations regarding nutritional strategies for athletes in both groups should be included.
Conclusions
The conclusions could be more detailed, with specific recommendations for nutritional interventions and the need for educating athletes.
The discussion and the conclusion have been modifyed adding the informations requested.
Reviewer 2 Report
Comments and Suggestions for Authors
This manuscript investigates the relationship between energy availability (EA) and body composition in endurance athletes and rugby players. While the topic is relevant and timely, several methodological limitations and inconsistencies in the presented data necessitate significant revisions before the manuscript is suitable for publication.
1. The study's small sample size, particularly within the endurance athlete group (n=18, with only 3 females), raises concerns about the generalizability of the findings. The imbalance in gender representation further limits the scope of the conclusions that can be drawn, particularly regarding female endurance athletes. A larger, more balanced sample is crucial for robust statistical analysis and broader applicability of the results.
2. The term "elite athlete" lacks a precise definition. The manuscript needs to clearly define the criteria used for classifying athletes as "elite," including specific performance metrics or competitive levels (e.g., national/international ranking, professional status). Furthermore, the characterization of training load is superficial. Simply stating weekly training hours is inadequate. Detailed information on training intensity, type (e.g., strength, speed, endurance), and specific training plans is essential to accurately estimate energy expenditure and contextualize EA within the training regime.
3. The manuscript introduces the concept of low energy availability (LEA) and categorizes it into clinical, subclinical, and optimal states, but fails to provide specific numerical thresholds for these classifications. Clear definitions are crucial for interpreting the results and comparing them with existing literature. The authors should explicitly state the kcal/kg FFM/day cutoffs used for each category.
4. The discussion section requires substantial strengthening. The rationale connecting higher body mass in rugby players to higher EA is underdeveloped. While increased caloric intake is necessary to maintain a larger physique, it doesn't inherently translate to higher EA. A more nuanced discussion exploring factors such as nutritional knowledge, dietary habits, and weight management strategies is needed. Furthermore, the observed differences in phase angle, while statistically significant, are within the normal range and their clinical significance requires further elaboration.
5. The conclusion section lacks depth and practical implications. The authors should expand on the significance of their findings and offer specific recommendations for practitioners and athletes based on the observed EA discrepancies between the two groups. This could include suggestions for nutritional interventions, monitoring strategies, and education initiatives to optimize EA and mitigate the risks associated with LEA.
6. The statement regarding ethical approval is insufficient. The manuscript claims that ethical approval was not necessary due to ongoing monitoring at the University of Florence Sports Medicine Centre. This is not a valid justification. All research involving human participants requires ethical review and approval from an appropriate institutional review board (IRB). The authors must provide details of IRB review and approval, including the IRB name and approval number, or clearly explain the process for obtaining a waiver if applicable. Additionally, the content of the informed consent process and data availability procedures need to be described in more detail.
7. The manuscript requires careful editing for grammar, clarity, and style. Several sentences are awkwardly phrased and could benefit from revision.
Author Response
This manuscript investigates the relationship between energy availability (EA) and body composition in endurance athletes and rugby players. While the topic is relevant and timely, several methodological limitations and inconsistencies in the presented data necessitate significant revisions before the manuscript is suitable for publication.
- The study's small sample size, particularly within the endurance athlete group (n=18, with only 3 females), raises concerns about the generalizability of the findings. The imbalance in gender representation further limits the scope of the conclusions that can be drawn, particularly regarding female endurance athletes. A larger, more balanced sample is crucial for robust statistical analysis and broader applicability of the results.
Thank you for supporting us in the improvement of the paper . These aspect have been described in the limit of the study . We consider this a pilot study , developed in an outpatient setting
- The term "elite athlete" lacks a precise definition. The manuscript needs to clearly define the criteria used for classifying athletes as "elite," including specific performance metrics or competitive levels (e.g., national/international ranking, professional status).
- Furthermore, the characterization of training load is superficial. Simply stating weekly training hours is inadequate. Detailed information on training intensity, type (e.g., strength, speed, endurance), and specific training plans is essential to accurately estimate energy expenditure and contextualize EA within the training regime.
Thank you for underline this crucial aspect . The athletes were not effective “elite” athletes ,however subjects regularly training to partecipate in the competitive games . No singular details of the intensity regimen of training has been evaluated , as consequence the aim of the study is to globally investigate and compare the categories. The terms of “elite” has been therefore removed and substituted with professionals that referred to athletes with natural talent, and competitive spirit. Professional Athletes are well disciplined when it comes to rigorous practice and training. This is the characteristics of the athlete’s investigated .
Regarding the caracterization of intensitybof training , the suggestions is important , however we have considered this aspect out of the aim of the main message of the study . In this term, after your revision, we have preferred to describe the population as professional and not elitè. The title has been modifyed.
- The manuscript introduces the concept of low energy availability (LEA) and categorizes it into clinical, subclinical, and optimal states, but fails to provide specific numerical thresholds for these classifications. Clear definitions are crucial for interpreting the results and comparing them with existing literature. The authors should explicitly state the kcal/kg FFM/day cutoffs used for each category.
The defitinion of LEA and the cut of for a correct interpretation, has been added
- The discussion section requires substantial strengthening. The rationale connecting higher body mass in rugby players to higher EA is underdeveloped. While increased caloric intake is necessary to maintain a larger physique, it doesn't inherently translate to higher EA. A more nuanced discussion exploring factors such as nutritional knowledge, dietary habits, and weight management strategies is needed. Furthermore, the observed differences in phase angle, while statistically significant, are within the normal range and their clinical significance requires further elaboration.
The potential proposal strategies for a correct nutritional support, have been discussed.
- The conclusion section lacks depth and practical implications. The authors should expand on the significance of their findings and offer specific recommendations for practitioners and athletes based on the observed EA discrepancies between the two groups. This could include suggestions for nutritional interventions, monitoring strategies, and education initiatives to optimize EA and mitigate the risks associated with LEA.
The conclusion has been implemented
7.The statement regarding ethical approval is insufficient. The manuscript claims that ethical approval was not necessary due to ongoing monitoring at the University of Florence Sports Medicine Centre. This is not a valid justification. All research involving human participants requires ethical review and approval from an appropriate institutional review board (IRB). The authors must provide details of IRB review and approval, including the IRB name and approval number, or clearly explain the process for obtaining a waiver if applicable. Additionally, the content of the informed consent process and data availability procedures need to be described in more detail.
This aspect has been clarifyed in the text. This not a protocol study and therefore no ethical committe is required . The data are collected during the regular access to perform the exame for the eligibility , that is in our center a complex evaluation .
- The manuscript requires careful editing for grammar, clarity, and style. Several sentences are awkwardly phrased and could benefit from revision.
Thank you for the effort to improve our paper .
Round 2
Reviewer 1 Report
Comments and Suggestions for Authors
Accept in present form
Reviewer 2 Report
Comments and Suggestions for Authors
The authors have adequately addressed my previous comments, and the manuscript is now suitable for publication.